# Mercury Chloride but Not Lead Acetate Causes Apoptotic Cell Death in Human Lung Fibroblast MRC5 Cells via Regulation of Cell Cycle Progression

**DOI:** 10.3390/ijms22052494

**Published:** 2021-03-02

**Authors:** Ji-Young Kim, Mi-Jin An, Geun-Seup Shin, Hyun-Min Lee, Mi Jin Kim, Chul-Hong Kim, Jung-Woong Kim

**Affiliations:** Department of Life Science, Chung-Ang University, Seoul 06974, Korea; jykim@cau.ac.kr (J.-Y.K.); dksalwls333@gmail.com (M.-J.A.); rjstjq89@naver.com (G.-S.S.); lhmscb@naver.com (H.-M.L.); mjkim831025@gmail.com (M.J.K.); 01031e70067@gmail.com (C.-H.K.)

**Keywords:** HgCl_2_, PbAc, heavy metal, apoptosis, cell cycle arrest, human lung fibroblast cell, MRC5

## Abstract

Heavy metals are important for various biological systems, but, in excess, they pose a serious risk to human health. Heavy metals are commonly used in consumer and industrial products. Despite the increasing evidence on the adverse effects of heavy metals, the detailed mechanisms underlying their action on lung cancer progression are still poorly understood. In the present study, we investigated whether heavy metals (mercury chloride and lead acetate) affect cell viability, cell cycle, and apoptotic cell death in human lung fibroblast MRC5 cells. The results showed that mercury chloride arrested the sub-G_1_ and G_2_/M phases by inducing cyclin B1 expression. In addition, the exposure to mercury chloride increased apoptosis through the activation of caspase-3. However, lead had no cytotoxic effects on human lung fibroblast MRC5 cells at low concentration. These findings demonstrated that mercury chloride affects the cytotoxicity of MRC5 cells by increasing cell cycle progression and apoptotic cell death.

## 1. Introduction

In biological systems, metals are essential components for maintaining biological functions, such as the catalytic versatility of many enzymes [1]. In contrast, certain heavy metals cause adverse health effects, even at very low concentrations. Heavy metals, including lead (Pb), mercury (Hg), cadmium (Cd), chromium (Cr), and arsenic (As), are widespread environmental pollutants [2]. Most of them are found both naturally and as an outcome of human activities; they are found in paints, vehicle emissions, and batteries. In addition, heavy metals are used in consumer products, such as children’s toys and jewelry [3]. Humans are usually exposed to heavy metals through inhalation, ingestion routes, and the skin. Heavy metals can pollute the air, water, and soil [4]. Their main characteristics are that they are not biodegradable and that they persist in the environment for long periods [5]. Despite the efforts made to reduce the usage of heavy metals, environmental contamination still remains a major concern and causes serious ecotoxicological problems. Human exposure to heavy metals can have numerous adverse effects, including dysfunction of the central nervous, cardiovascular, endocrine, and immune systems [6,7,8].

The environmental emission of lead is mainly due to fossil fuel combustion and waste disposal and results in its accumulation in the upper soil layer [9]. Since lead is not degraded and is strongly absorbed into the soil, it continuously distributes in the environment. Lead is well studied with regard to its toxicity in various human organs. For example, lead passes to infants via lactation from exposed mothers; lead from the skeletal stores of the mothers is released into the blood and breast milk [10]. Lead is highly toxic to the central nervous system. Symptoms such as headache, attention span deficit, and loss of memory usually appear within 30 min of acute lead exposure [11]. Furthermore, several studies have shown that exposure to lead increases the occurrence of renal tumor, brain, lung, and hematopoietic cancer in rodents [12]. Mercury is used for numerous purposes and industries, including nuclear reactors, caustic soda production, electrical industry, and dentistry [13]. Humans are exposed to mercury via environmental pollution, dental care, occupational operations, and food contamination [14]. Because elemental vapor is highly lipophilic, it is rapidly absorbed into the lungs, blood–brain barrier, and placental barrier [15]. Exposure to mercury results in neurotoxicity, gastrointestinal toxicity, and nephrotoxicity [13]. Even though several studies have shown the adverse effects of lead and mercury in numerous cancers, it is still poorly understood whether heavy metals affect the physiology of human lung fibroblasts.

Lung tissue is the most common target of lead- and mercury-induced toxicity, and they cause lung cancer in human by inhalation. The toxic mechanisms contributed to mercury and lead include DNA damage repair processes, repression of tumor suppressor gene expression, and enzymatic activities in oxidative stress [16,17]. Mercury and lead elevate ROS levels by decrease of cellular glutathione (GSH) and antioxidants, disturb cellular signaling [17]. Moreover, toxic metals including mercury and lead have been found to act mimic essential metals and access to molecular targets resulting in apoptotic cell death, cell cycle arrest, and carcinogenesis [18,19]. Even if several studies have shown the adverse effect of lead and mercury in lung cancers, it still poorly understood whether mercury and lead affect the physiological changes of the human lung fibroblast.

It was the aim of the study to investigate the cytotoxicity of mercury chloride (HgCl_2_) and lead acetate (PbAc) on the molecular mechanism of human lung fibroblast. We hypothesized that HgCl_2_ and PbAc have different influences on cell viability, cell cycle, and apoptosis. Therefore, we used the human lung fibroblast cell line, MRC5, stimulated these cells with (6.25–50 μM) and PbAc (20–100 μM), and analyzed the cell viability properties, cell cycle by measuring the expression of cyclin B1 and D1, and apoptotic cell death through the caspase-3 pathway.

## 2. Results

### 2.1. Effect of Heavy Metals on Viability of Human Lung Fibroblast Cells

To determine the cytotoxicity of HgCl_2_ and PbAc on human lung fibroblast MRC5 cells, we first measured the cell viability of MRC5 cells using a MTS assay. The cells were grown with various concentrations of HgCl_2_ (6.25–50 μM) and PbAc (20–100 μM) for 24 and 48 h. The exposure to 50 μM HgCl_2_ induced cell death for both treatment durations, whereas the treatment with PbAc did not significantly affect cell viability (Figure 1A). The results showed that there are different sensitivities of MRC5 cells to heavy metals. Based on these results, for subsequent experiments, we chose the following concentration ranges: HgCl_2_ (12.5 and 50 μM) and PbAc (40 and 100 μM). To further demonstrate whether the exposure to HgCl_2_ and PbAc causes changes in cell morphology, we observed the cellular shape using phase-contrast microscopy. Notably, HgCl_2_ increased the number of abnormal morphology changes observed, such as dying cells, including cellular shrinkage, decrease in contact with neighboring cells, and formation of apoptotic cell bodies in a time- and dose-dependent manner (Figure 1B, left panel, and Appendix A). However, the cellular morphology of MCR5 cells did not change following PbAc treatment (Figure 1B, right panel). Furthermore, the cell viability was assessed in HgCl_2_- and PbAc-treated MRC5 cells using a live/dead assay. Consistent with the MTS assay and morphological changes results, the treatment with HgCl_2_ decreased the total cell number, as well as the number of dead cells, whereas PbAc had no significant effect on MRC5 cell viability (Figure 1C, and Appendix A). These results indicated that HgCl_2_ and PbAc had different cytotoxicity effects on human lung fibroblast MCR5 cells.

### 2.2. HgCl_2_ Treatment, but Not PbAc Treatment, Decreased MRC5 Cell Proliferation

Proliferating cells express high amounts of Ki-67, which is a well-known cell proliferation biomarker that can be used to detect the proportion of dividing cells [20]. We investigated whether HgCl_2_ or PbAc affects the proliferation of human lung fibroblast cells by measuring Ki-67 expression levels using flow cytometry analysis. After the cells were treated with different concentrations of HgCl_2_ or PbAc for 24 or 48 h, we immunostained them with Ki-67 antibody. The expression level of Ki-67 was slightly decreased upon HgCl_2_ treatment for 24 h, but this difference was recovered after a longer exposure (Figure 2A,B and Appendix A). In PbAc treatment, the proportion of Ki-67 expressing cells did not change (Figure 2A,B and Appendix A).

### 2.3. Effects of Heavy Metals on Cell Cycle Progression in Human Lung Fibroblast Cells

To further demonstrate the inhibitory effects of heavy metals on cell proliferation, human lung fibroblast MRC5 cells were treated with HgCl_2_ and PbAc for the indicated time periods. The group treated with 100 μM HgCl_2_ showed slightly increased proportion of cells in the sub-G_1_ phase (5%) compared to the control group (0.5%) in a dose-dependent manner (Figure 3A,B). In addition, the number of cells in the G_2_/M phase increased in HgCl_2_ treated-MRC5 cells. In contrast, the number of cells in the G_0_/G_1_ phase was significantly decreased after HgCl_2_ treatment (Figure 3A,B and Appendix A). Consistent with the cell proliferation assay results, PbAc treatment did not affect cell cycle progression (Figure 3A,B and Appendix A). These results show that HgCl_2_ and PbAc differentially regulate cell cycle arrest in human lung fibroblast MRC5 cells.

### 2.4. Cyclin B1 and Cyclin D1 Expression in Heavy Metal-Treated MRC5 Cells

Since HgCl_2_ increased the number of cells in the sub-G_1_ phase and caused cell cycle arrest, we investigated the detailed molecular mechanisms of cell cycle progression. MRC5 cells were treated with HgCl_2_ and PbAc at different concentrations for 24 or 48 h, and were immunostained with anti-cyclin B1 and cyclin D1 antibodies. Cyclin B1 or cyclin D1 expressing cells were counted using FACS. The cyclin B1-positive population, denoting the G_0_/G_1_ phase, was significantly increased to 24% at 24 h, and 20% at 48 h in HgCl_2_-treated cells (Figure 4A,B and Appendix A). In addition, the cyclin D1-positive population, representing the G_2_/M phase, was slightly increased, but the range of change was insignificant (Figure 5A,B and Appendix A). We also detected cyclin B1 and cyclin D1 after PbAc treatment in MRC5 cells. As expected, the expression of cyclin B1 or cyclin D1 did not show significant changes following PbAc treatment (Figure 4 and Figure 5, and Appendix A). These results indicate that the accumulation of cyclin B1 leads to cell cycle arrest at the sub-G_1_ and G_2_/M phases in MRC5 cells treated with HgCl_2_.

### 2.5. Effect of Heavy Metals on Apoptotic Cell Death in MRC5 Cells

Cellular proliferation is regulated by two major events: cell cycle progression and apoptotic cell death. Having demonstrated the cell cycle arrest at the sub-G_1_ phase in HgCl_2_-treated human lung carcinoma cells, we next investigated whether HgCl_2_ and PbAc are associated with apoptotic cell death. We double-stained the cells with annexin V and PI to analyze the apoptotic population in HgCl_2_- or PbAc-treated MRC5 cells using flow cytometric analysis. Compared to the control group, the exposure to HgCl_2_ significantly induced early and late apoptotic cell death (Figure 6A,B). The annexin-V- and PI-double-positive populations were approximately four- to sixfold higher, respectively, at the final concentration of HgCl_2_ treatment (50 μM) in a dose- and time-dependent manner (Figure 6A,B and Appendix A). In contrast, the annexin-V- and PI-positive cell population was not significantly different between EtOH- and PbAc-treated MRC5 cells (Figure 6A,B and Appendix A).

Recent studies have shown that heavy metals induce apoptosis via caspase activation [21]. To further investigate the molecular mechanism of apoptosis induced by HgCl_2_ treatment, we detected cleaved caspase-3, which is the active form of caspase-3. The cleaved caspase-3 population doubled after HgCl_2_ treatment for 24 or 48 h (Figure 7A,B and Appendix A), whereas there was no significant difference in PbAc-treated MRC5 cells (Figure 7A,B and Appendix A). These data suggest that heavy metals, especially HgCl_2_, cause apoptotic cell death via the induction of caspase-3 activity in human lung fibroblast cells.

## 3. Discussion

In the present study, we investigated the adverse effects of heavy metals on human lung fibroblast cells. Even if heavy metals are well studied in various cancers, it is unclear whether mercury and lead have carcinogenetic or cytotoxic effects in lung cancer. Therefore, we determined whether mercury and lead affect the viability of the human lung fibroblast MRC5 cell line. We found that mercury decreased cell proliferation via cell cycle arrest and apoptosis. However, lead had no significant effect on cell viability. These results indicate that mercury induces cytotoxicity, whereas lead at low concentrations have no adverse effects on human lung fibroblast cells.

The main sources of Pb exposure are drinking water, food, cigarettes, and industrial sources, such as gasoline, plumbing pipes, house paint, storage batteries, and children’s toys [22]. Adults are exposed to 30–50% of lead via drinking water, and children may have a 50% higher lead exposure. Mercury is found in batteries, dental amalgams, and thermometers. In addition, it is used in some electrical equipment, automotive, and building industries. Mercury can be released into the air, which is then inhaled by humans. The Joint FAO/WHO Expert Committee on Food Additives (JECFA) determined that the tolerable absorption amount of lead and mercury was 3.5 µg/kg and 0.2 µg/kg body weight per day, respectively [23,24]. Lead and mercury are absorbed and reach the kidneys, followed by the heart, brain, liver, and bones in the body. In particular, exposure of pregnant women to lead influences baby growth by reducing the birth weight and inducing neurodevelopmental abnormalities [25].

Most heavy metals have carcinogenic or cytotoxic effects. Heavy metals target several cellular regulatory proteins or signaling proteins that are involved in apoptosis, DNA repair, DNA methylation, cell cycle progression, and cell proliferation [26]. Moreover, certain heavy metals induce carcinogenesis by activating redox-related transcription factors, including p53, Ap-1 and NF-kB via reprocessing of electrons via antioxidant signaling. These transcription factors regulate expression of genes associated with apoptotic cell death, cell cycle, and proliferation [27]. Lead-induced carcinogenesis involves induction of DNA damage, alterations in DNA repair system, and the generation of ROS [12]. Several studies have supported that ROS generation due to lead exposure alters the chromosomal structure and transcription by replacing zinc ion [12]. Mercury also has several health effects in humans. The exposure to mercury induces the generation of free radicals, thereby increasing oxidative stress, damaging DNA, proteins, and lipids [28].

Ki-67 protein has been known as a proliferation marker for human tumor cells and is widely used in a pathological investigation. Ki-67 is strongly expressed during all active phases of the cell cycle (G_1_, S, G_2_ and mitosis), but downregulated in G_0_ phase cells [29]. Since the Ki-67 is present in all cycling cells, it would be evident that the expression of Ki-67 is an excellent proliferating marker in the cell population. As expected, we observed that the exposure to Hg decreased Ki-67 expression level in MRC5 cells at 24 h treatment (Figure 2). Furthermore, we determined the effects of heavy metals on cell proliferation (Figure 3, Figure 4 and Figure 5). The cell cycle is controlled by different cellular proteins. The cyclin-dependent kinases (CDK) and their activating proteins (cyclins) are key regulatory proteins that are activated at specific points of the cell cycle [30]. CDK protein levels are stably maintained during the cell cycle, but cyclin protein expressions increase and decrease during cell cycle progression. The cyclin D1 associates with CDK4 and CDK6 regulates progression from G1 into S phase. The cyclin B1 expression increases in G2 phase, and controls of cell cycle progression from G2 to M phase by interacting with CDK1 [31]. Our results showed that the cell cycle is arrested at sub-G1 and G2/M phases by Hg treatment (Figure 3). It might be caused by the accumulation of cyclin B1 in MRC5 cells treated with Hg (Figure 4 and Figure 5).

Caspase-3 belongs to a cysteine protease family which are crucial role for apoptosis, regulation programmed cell death, and maintenance of cell homeostasis [32]. There are two main apoptosis pathways including the extrinsic and the intrinsic pathways. The extrinsic pathway is regulated by the binding of ligands to their receptors (TNFα, FasL, and TRAIL). The mitochondrial pathway (intrinsic) is activated by DNA or cellular damage [33]. Although the induction of ROS generation by heavy metals was not elucidated in this study, our results showed that Hg increased apoptotic cell death in human lung fibroblast MRC5 cells (Figure 6 and Figure 7). One potential mechanism underlying of cytotoxicity in human fibroblast cell growth is alterations in apoptosis. Since the activation of caspase-8 (extrinsic) and capase-9 (intrinsic) induces the alteration in caspase-3 activity, it might be one of the best experiments to explain apoptosis pathway by Hg exposure in human lung fibroblast MRC5 cells. Previous studies have shown that Hg has damaging effects on mitochondria in cerebellar granule cells [34]. In addition, mercury activates the caspase-3 pathway, which is the final apoptotic signaling pathway in differentiated human neurons, astrocytes, and the cortex of adult rats [35,36]. Consistent with this evidence, our study also proposed that mercury induced apoptosis via an increase in caspase-3 activity.

The mechanism of lead-induced cytotoxicity is also well-known. For instance, lead arrests the cell cycle in the G_0_/G_1_ phase, followed by apoptotic cell death of leukemia cells [37]. The long-term exposure to lead induced cytotoxicity in hippocampal HT-22 cells [38]. However, our results indicate that lead at a low concentration and for a short exposure time has no significant cytotoxic effect on human lung fibroblast cells. Based on the cell viability assay, we decided to use a lead concentration of 20–100 μM for 24 or 48 h, which showed no effect on cell growth. Because there is little known about the mechanisms underlying the cytotoxicity in lead- and mercury-treated human lung fibroblast cells, we still need more experiments with high concentration and long-term exposure. Furthermore, to further investigate each or comprehensive effects of mercury and lead, we need to perform genome-wide analysis including mRNA-seq, DNA methylation sequencing. It would be helpful to perceive the adverse effects of chemical derivatives and/or analogues in various cell lines in the future studies.

## 4. Materials and Methods

### 4.1. Chemicals and Reagents

Mercury chloride (HgCl_2_; cat. no. 316512) and Lead acetate (PbAc; cat. no. 215465) were purchased from Sigma-Aldrich (Oakville, ON, Canada). A 1M stock solution of HgCl_2_ and PbAc were prepared and diluted in 100% ethanol (EtOH). We used 6.25, 12.5, 25, and 50 μM of HgCl_2_ and 20, 40, 80, and 100 μM of PbAc in Dulbecco′s Modified Eagle′s Medium (DMEM) containing 10% fetal bovine serum (FBS). MTS assay kit (cat. no. G3581, Madison, WI, USA) was purchased from Promega, and the LIVE/DEAD kit (cat. no. L3224, CA, USA) was purchased from Invitrogen. Apoptosis Detection kit (Cat. No. 556547, CA, USA) was purchased from BD Pharmingen^TM^. The Ki-67 antibody (Cat. No. ab15580), cyclins B1 (Cat. No. ab32053, CA, USA), cyclin D1 (Cat. No. 2978S, Beverly, MA, USA) were obtained from Abcam, Santa Cruz Biotechnology, Cell Signaling, respectively. The secondary goat antirabbit antibody was purchased from Jackson ImmunoResearch (Cat. No. 111-095-144, West Grove, PA, USA).

### 4.2. Cell Culture

All reagents for cell culture were obtained from Welgene (Seoul, Korea). The human lung fibroblast cell line, MRC5, was purchased from the Korean Cell Line Bank (KCLB) (Seoul, Korea). MRC5 cells were grown in Dulbecco’s Modified Eagle’s Medium (DMEM) supplemented with 10% FBS and 1% penicillin/streptomycin at 37 °C in an incubator in an atmosphere of 5% CO_2_. At first, 0.1 mg/mL of poly-D lysine was placed on coverslips for 6 h at 23 °C, then MRC5 cells were cultured in 100 mm culture dishes at a density of 2.5 × 10^6^ cells/dish. After 24 h, various concentration of HgCl_2_ and PbAc was treated in MRC5 cells for 24 h and 48 h. The cells on coverslips were used for the Live/Dead cell assay, and the cells in the 10 cm culture dishes were processed for further experimental analyses. For each assay, 0.6% EtOH was used as the negative control.

### 4.3. MTS Assay

MRC5 cells were cultured at a density of 2 × 10^4^ cells/well in 48-well plates and were subsequently treated with different concentrations of HgCl_2_ and PbAc in 100 μL DMEM supplemented with 10% FBS for 24 or 48 h. The MRC5 cells were treated with 10% MTS (3-(4,5-dimethylthiazol-2-yl)-5-(3-carboxymethoxyphenyl)-2-(4-sulfophenyl)-2H-tetrazolium) and incubated for 1 h at 37 °C. The MTS assay protocol is based on the reduction of the MTS tetrazolium compound by viable mammalian cells (and cells from other species) to generate a colored formazan dye that is soluble in cell culture media. This conversion is thought to be carried out by NAD(P)H-dependent dehydrogenase enzymes in metabolically active cells. The formazan dye is quantified by measuring the absorbance at 490 nm using a Multiskan GO microplate reader 323 (Waltham, MA, USA).

### 4.4. Morphological Change and Live/Dead Cell Assay

MRC5 cells were cultured on coated coverslips and treated with different concentrations of HgCl_2_ and PbAc for 24 and 48 h. The morphological changes were observed using an inverted microscope. For live/dead cell the LIVE/DEAD^®^ Cell Imaging Kit, live cells are distinguished by the presence of ubiquitous intracellular esterase activity as determined by the enzymatic conversion of the virtually nonfluorescent cell-permeant calcein AM to the intensely fluorescent calcein, which is well-retained within live cells. The red component of the LIVE/DEAD^®^ Cell Imaging Kit is cell-impermeant and therefore only enters cells with damaged membranes. In dying and dead cells, a bright red fluorescence is generated upon binding to DNA. Background fluorescence levels are inherently low with this assay technique because the dyes are virtually nonfluorescent before interacting with cells. The fluorophores in the LIVE/DEAD^®^ Cell Imaging Kit were selected for their brightness, spectral properties (FITC and Texas Red filters), and ease of use. MRC5 cells were double-stained with 3 μM Ethidium Homodimer-1 (EthD-1) and 0.3 μM calcein-AM mixture (Life Technologies, CA, USA) for 30 min in the dark condition. The morphology of the unwashed cells was observed using Nikon Eclipse TE300 Inverted Fluorescence Microscope (Nikon Corp., Tokyo, Japan).

### 4.5. Cell Cycle Analysis

MRC5 cells were collected and were fixed in 70% ethanol with rotator for 1 h at 4 °C. Cell pellets were resuspended in 1X PBS containing 0.25 μg/μL RNase A and incubated for 1 h at 37 °C. The cells were treated with 10 μg/mL PI and incubated for 15 min in the dark at 23 °C. Finally, the cells were added 300 μL 1X PBS and then were analyzed using a BD Accuri™ C6 Plus flow cytometer (BD FACS, Franklin Lakes, NJ, USA). A minimum of 10,000 cells were considered per sample and the results are represented as histograms for analyzing cell distribution in the different phases of the cell cycle. The cell cycle profile was analyzed using BD Accuri™ C6 Plus software (BD FACS, Franklin Lakes, NJ, USA).

### 4.6. Annexin V-FITC/Propidium Iodide (PI) Apoptosis Assay

A FITC Annexin-V Apoptosis Detection Kit I (BD pharmingenTM, La Jolla, CA, USA) was used for determining cellular apoptotic cell death according to the manufacturer’s instructions. Briefly, cells were collected and resuspended in 1X Annexin-V binding buffer (140 mM NaCl, 2.5 mM CaCl_2_, and 10 mM HEPES/NaOH (pH 7.4)). Next, the cells were double-stained with 5 μL PI and/or 5 μL annexin-V Alexa Fluor 488 and incubated for 15 min in the dark condition and were washed with 1X annexin-V binding buffer. At least 10,000 cells were considered per sample and apoptosis was quantified by the BD Accuri™ C6 Plus flow cytometer (BD FACS, Franklin Lakes, NJ, USA).

### 4.7. Immunostaining for Fluorescence-Activated Cell Sorting (FACS) Analysis

Heavy metals-treated MRC5 cells were fixed in 1% paraformaldehyde for 5 h on a rotator at 4 °C and centrifuged at 3000 rpm for 3 min. Each sample was resuspended with solution A (75 mM sodium acetate, 0.1% saponin, 0.1% BSA, and 25 mM HEPES, pH 7.2) and was incubated in the diluted indicated antibodies against cyclin B1 (1: 200), cyclin D1 (1:200), Ki-67 (1:400), and cleaved capase-3 (1:200) for 1 h at 23 °C. After washing with 1X PBS, cells were incubated with FITC-labeled goat antirabbit secondary antibody (1:200) for 30 min in the dark at 23°C. The expression of the target proteins was quantified using a BD Accuri™ C6 Plus flow cytometer (BD FACS, Franklin Lakes, NJ, USA). A minimum of 10,000 cells were considered per sample and the results are represented as histograms. The percentage of the cell population was analyzed using BD Accuri™ C6 Plus software (BD FACS, Franklin Lakes, NJ, USA).

### 4.8. Statistical Analyses

Data are presented as the mean ± standard error of the mean (S.E.M); experiments were performed in triplicate. Data were analyzed using two-way ANOVA followed by Tukey’s multiple comparison test using GraphPad Prism5 software (San Diego, CA, USA). Differences between the groups were considered to be significant at *p* < 0.05.

## 5. Conclusions

This study is the first, to our knowledge, to suggest the mechanisms underlying the possible effects of mercury and lead on human lung fibroblast cells. We examined basic and crucial experiments for cell viability and proliferation of MRC5 cells. We demonstrated that mercury reduced cell viability by increasing cell cycle arrest and apoptotic cell death. On the other hand, lead did not affect the normal development of human lung fibroblast cells. Following recent studies, diverse environmental factors can alter gene expression and epigenetic regulation. Therefore, future studies will help to determine whether heavy metals modulate the expression of cell viability-related genes in human lung fibroblast cells.

## Figures and Tables

**Figure 1 ijms-22-02494-f001:**
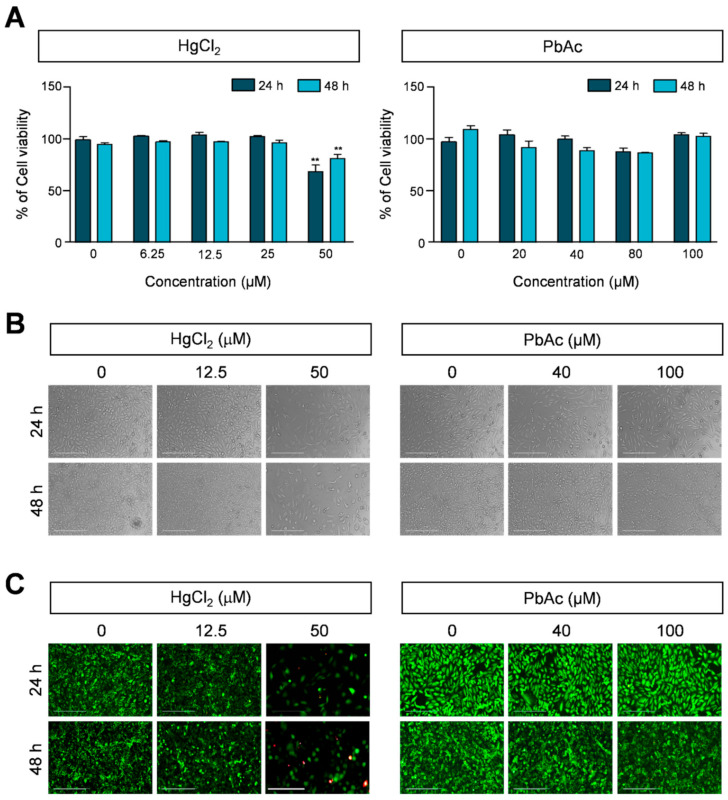
Effect of heavy metals on cell viability and morphology of MRC5 cells. (**A**) MRC5 cell viability was measured using the MTS assay. Cells were treated with 6.25–50 μM of HgCl_2_ and 20–100 μM PbAc for 24 or 48 h. Data are represented as the percentage of the values obtained for the EtOH-treated cells used as the control. Error bars show mean ± SEM. *p* values obtained by Student’s *t*-test. ** *p* value < 0.05. (**B**) The morphology of MRC5 cells was observed by phase-contrast microscopy after treatment with indicated concentration of HgCl_2_ and PbAc for 24 or 48 h. (**C**) MRC5 cells were stained with calcein-AM (green) and ethidium homodimer (red) by the live/dead assay. EtOH were used as the negative control. Images are representative of three independent experiments. Scale bars represent 200 μM.

**Figure 2 ijms-22-02494-f002:**
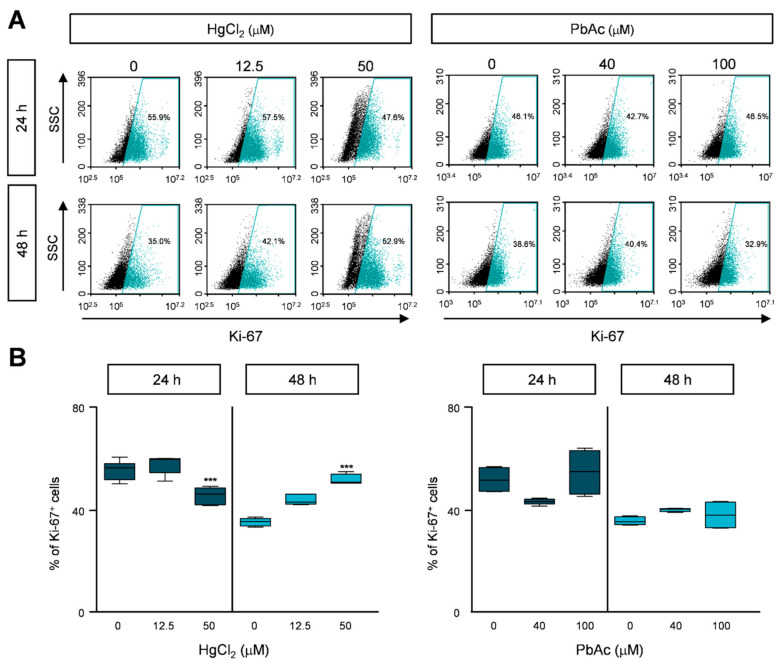
Heavy metals treatment reduced the proliferation of MRC5 cells. (**A**) The heavy metals-treated MRC5 cells were immunostained with anti-Ki-67 antibodies. The cells were counted by FACS analysis. (**B**) The percentages of Ki-67-positive population are represented as the mean ± S.E.M. of three independent experiments (n = 6), each performed in triplicate. Error bars show mean ± S.E.M. *p* values obtained by Student’s *t*-test. *** *p* value < 0.001.

**Figure 3 ijms-22-02494-f003:**
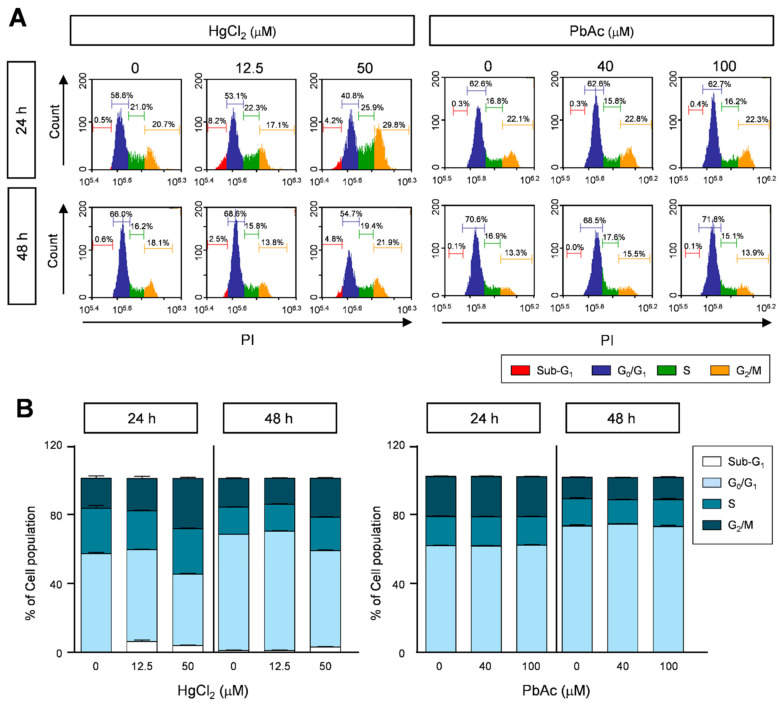
Cell cycle progression in heavy metals treatment in MRC5 cells. (**A**) The heavy metals-treated MRC5 cells were stained with propidium iodide (PI), and the cell cycle was measured using FACS analysis. (**B**) The percentages of population in the sub-G_1_, G_0_/G_1_, S, and G_2_/M phases are represented as the mean ± S.E.M. of three independent experiments (n = 6), each performed in triplicate.

**Figure 4 ijms-22-02494-f004:**
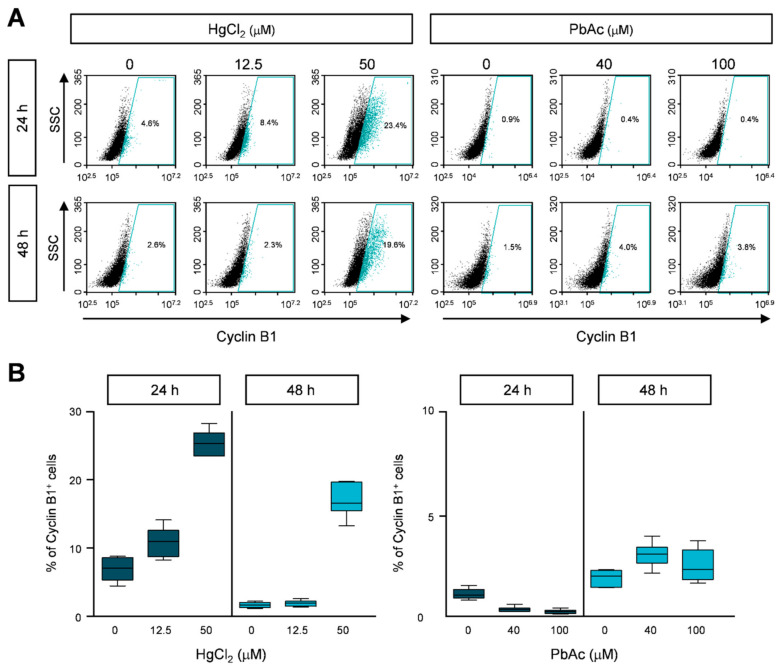
The effect of heavy metals on cyclin B1 expression in MRC5 cells. (**A**) Cells were treated with indicated concentration of heavy metals for 24 or 48 h. Cells were fixed with 1% PFA and stained with anti-cyclin B1 antibody. The cyclin B1 expression was analyzed by FACS analysis. (**B**) The percentages of cyclin B1-positive cells are represented as the mean ± S.E.M. of three independent experiments (n = 6), each performed in triplicate. EtOH was used as the negative control.

**Figure 5 ijms-22-02494-f005:**
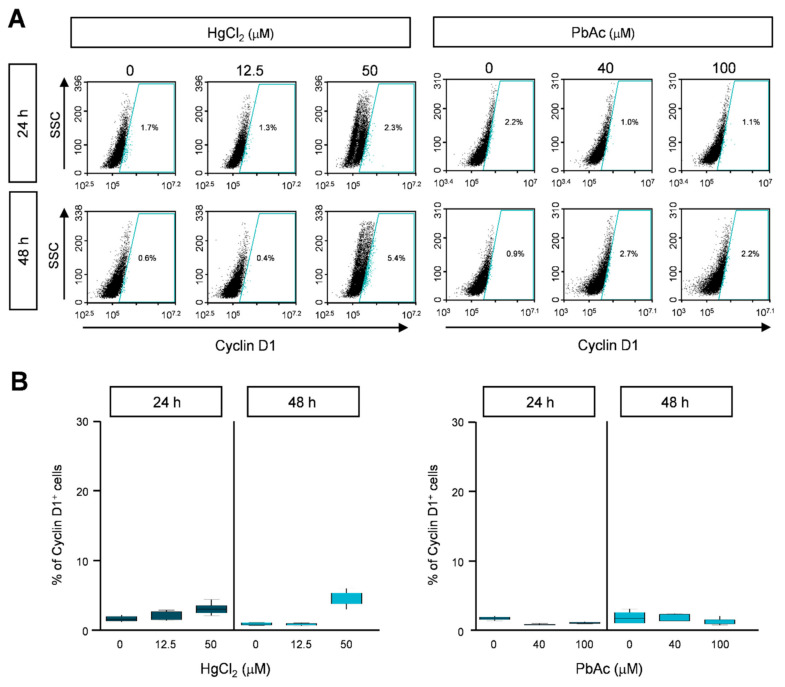
The effect of heavy metals on cyclin D1 expression in MRC5 cells. (**A**) Cells were treated with indicated concentration of heavy metals for 24 or 48 h. Cells were fixed with 1% PFA and stained with anti-cyclin D1 antibody. The cyclin D1 expression was analyzed by FACS analysis. (**B**) The percentages of cyclin D1-positive cells are represented as the mean ± S.E.M. of three independent experiments (n = 6), each performed in triplicate. EtOH was used as the negative control.

**Figure 6 ijms-22-02494-f006:**
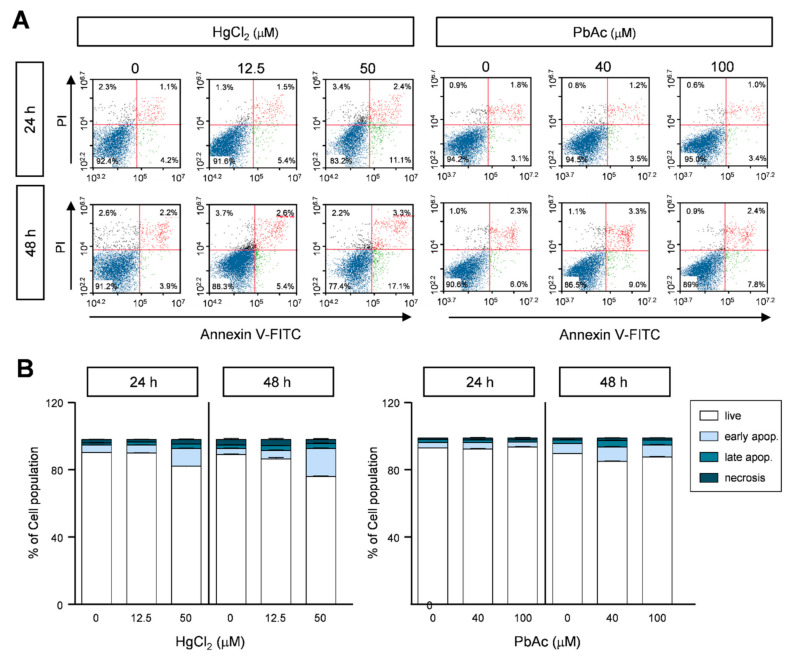
Effect of apoptotic cell death upon heavy metals treatment in MRC5 cells. (**A**) Cells were treated with indicated concentration of heavy metals for 24 or 48 h, and were double-stained with annexin V-FITC and PI. The proportion of apoptotic cells was assessed by FACS analysis. The scatter plots represent PI (Y-axis) and annexin V-FITC (X-axis). (**B**) The percentages of cells in the live, early- and late-apoptotic, and necrotic stages are expressed as the mean ± S.E.M. of three independent experiments (n = 6), each performed in triplicate.

**Figure 7 ijms-22-02494-f007:**
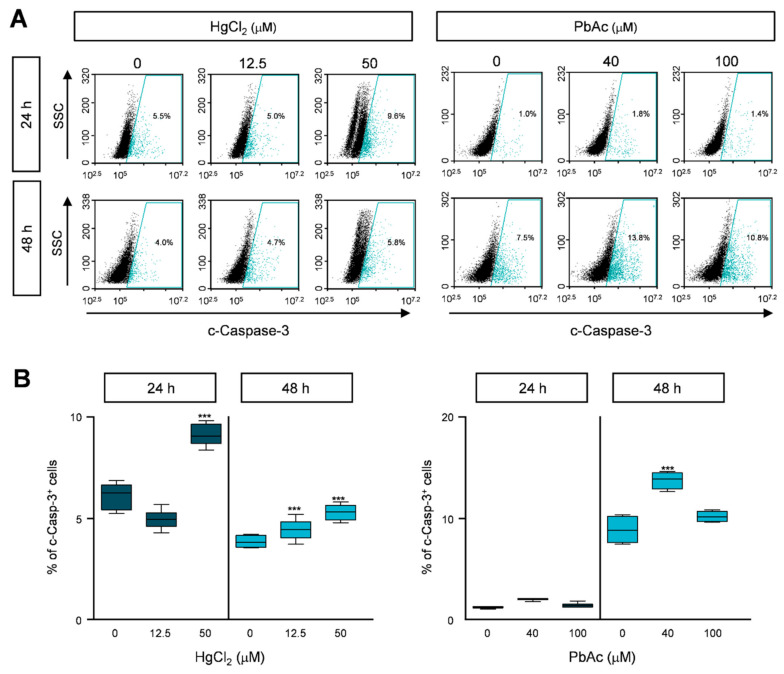
The effect of caspase-3 activity in heavy metals-treated MRC5 cells. (**A**) Cells were treated with indicated concentration of heavy metals for 24 or 48 h. Cells were fixed with 1% PFA and stained with anti-cleaved caspase-3 antibody. Expression levels of cleaved caspase-3 were analyzed by FACS analysis. (**B**) The percentages of cleaved caspase-3-positive cells are represented as the mean ± S.E.M. of three independent experiments (n = 6), each performed in triplicate. EtOH was used as the negative control. Error bars show mean ± SEM. *p* values obtained by Student’s *t*-test. *** *p* value < 0.001.

## Data Availability

The data presented in this study are available in the article or Appendix A.

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
