# Peer review of "Mercury Chloride but Not Lead Acetate Causes Apoptotic Cell Death in Human Lung Fibroblast MRC5 Cells via Regulation of Cell Cycle Progression"

_ijms, 2021, doi:10.3390/ijms22052494_

Round 1

Reviewer 1 Report

  1. I strongly recommend English native speaker correction. Many sentences needs a verb, or need to be rearranged.
  2. I the main title I would use normal words not abbreviations i.e. mercury chloride, lead acetate (abbrev. Ac in main title is not clear for me)
  3. Abstract needs to be more clear. Authors too long write about the metals toxicity in this part. Abstract should be more concise. The methods in abstract are missed.
  4. Introduction: please chcracterise the state of knowledge about the molecular pathways od metal influence on lungs. This part Authors treated too widely, and should focus significantly more attention on molecular aspects, especially in lungs. Authors wrote many basis data that needs be rewritten to the basic, and molecular aspects needs to be presented. 
  5. Aim of study should be significantly pointed at the end of Introduction. Is almost missed, and not justified - arose from almost nothing.
  6. Line 272 - please explain DMEM
  7. Line 274 - MTS kit, what is it, and why is used for? The same for Leave/Dead, this reagent shoud be introduces, whay was used for, how it works.
  8. All these kits in the first paragraph of Methods should be characterized why were used for.
  9. Authors worked on human cell lines - I do not see any agreements for such research.
  10. In my opinion cited literature is not actual, please cite works from last ten years.
  11. How used metals ranges correspond to the environmentsl exposure. Why Authors used such metal concenrations? Is the aim was to detect anormalities, if yes the critic dose was estimated?
  12. Why mercury and lead were used in these particular forms?
  13. Line 183 - Authors wrote "recent studies" and gave only 1 reference, please add more ones.
  14. "Even if heavy metals are well studied in various cancers, it is unclear
    194 whether heavy metals have carcinogenetic or cytotoxic effects in lung cancer." - very not informative. Authors study 2 metals not all of them.
  15. Figure 6 and 7 must be in results, and when we will do this, we will see that a duscussion part is almost absent. Results that authors received needs to be earnestly discussed with the most actual results involved in most recent literature. Lots od sentences in discussion chapter overlap with very general and not precise sentences from Introduction chapter.
  16. References needs to be enriched with the significant papers to this study.

Author Response

[Comments and Suggestions for Authors]

We all appreciate for the reviewer found the importance and critiques from our study of the functional effects of heavy metal in MRC5 cells. We carefully read reviewer’s suggestions and tried to answer in a logical manner to improve the quality of this manuscript. Changes are easily visualized red color in the revised manuscript.

  • I strongly recommend English native speaker correction. Many sentences needs a verb, or need to be rearranged.

➢ We thank the reviewer for identifying the error in previously submitted manuscript. We changed and modified the manuscript as reviewer mentioned. Also, we have rephrased the manuscript and gotten second language corrections from professional editors. We also included the certificate of language editing in the cover letter.

  • [the main title] would use normal words not abbreviations i.e. mercury chloride, lead acetate (abbrev. Ac in main title is not clear for me)

➢ As reviewer’s suggestion, we changed the main title to “Mercury Chloride and Lead Acetate cause apoptotic cell death in human lung fibroblast cells via regulation of cell cycle pro-gression”

  • Abstract needs to be more clear. Authors too long write about the metals toxicity in this part. Abstract should be more concise. The methods in abstract are missed.

➢ As reviewer mentioned, we modified abstract section as below. We rearranged the part of metals toxicity contents, and we rewrote the additional information of results in our present study. That added paragraph was marked as red text.

➢ “Heavy metals are important for various biological systems, but, in excess, they pose a serious risk to human health. Heavy metals are commonly used in consumer and industrial products. Despite the increasing evidence on the adverse effects of heavy metals, the detailed mechanisms underlying their action on lung cancer progression are still poorly understood. In the present study, we inves-tigated whether heavy metals affect cell viability, cell cycle, and apoptotic cell death in human lung fibroblast MRC5 cells. The results showed that mercury chloride arrested the sub-G1 and G2/M phases by inducing cyclin B1 expression. In addition, the exposure to mercury chloride increased apoptosis through the activation of caspase-3. However, lead had no cytotoxic effects on MRC5 cells at low concetrantion. These findings demonstrated that heavy metals, especially mercury chloride, affect the cytotoxicity of MRC5 cells by increasing cell cycle progression and apoptotic cell death.”

  • Introduction: please chcracterise the state of knowledge about the molecular pathways of metal influence on lungs. This part Authors treated too widely, and should focus significantly more attention on molecular aspects, especially in lungs. Authors wrote many basis data that needs be rewritten to the basic, and molecular aspects needs to be presented.

➢ We appreciate for reviewer’s suggestions. We rearranged introduction part and added the mechanism of mercury and lead on human lung cells as below. That added paragraph was marked as red text.

➢ “The Lung tissue is most common targets of lead and mercury-induced toxicity, and they cause lung cancer in human by inhalation. The toxic mechanisms contributed to mercury and lead include DNA damage repair processes, repression of tumor suppressor gene expression, and enzymatic activities in oxidative stress. Mercury and lead elevate ROS levels by decrease of cellular glutathione (GSH) and antioxidants, disturb cellular signaling. Moreover, toxic metals including mercury and lead have been found to act mimic essential metals and access to molecular targets resulting in apoptotic cell death, cell cycle arrest, and carcinogenesis. Even if several studies have shown the adverse effect of lead and mercury in lung cancers, it still poorly understood whether mercury and lead affect the physiological changes of the human lung fibroblast.

  • Aim of study should be significantly pointed at the end of Introduction. Is almost missed, and not justified - arose from almost nothing.

➢ As reviewer’s suggestion, we rewrote the aim of this study in the introduction section as below. That added paragraph was marked as red text.

➢ “In the present study, we demonstrated the cytotoxicity of heavy metals by using a human lung fibroblast cell line, MRC5. We investigated the molecular mechanisms by which heavy metals affected the cellular progress of the human lung fibroblast cells. We found that mercury chloride (HgCl2) significantly decrease cell viability of human lung fibroblast cells. In addition, we examined that HgCl2 increased cell cycle arrest at sub-G1 and G2/M phase. Furthermore, the apoptotic cell death was increased via caspase-3 activation in HgCl2 treated MRC5 cells. However, lead acetate (PbAc) has no effect on the development of human lung fibroblast cells. These results indicate that mercury chloride leads to the cell cycle arrest and apoptotic cell death through caspase-3 pathway in human lung fibroblast MRC5 cells.”

  • Line 272 - please explain DMEM

➢ We changed to Dulbecco’s Modified Eagle’s Medium (DMEM). DMEM is the most broadly suitable medium for many adherent cell phenotypes among defined media for cell and tissue culture. The Dulbecco’s modification is an enhanced supplementary formulation that boosts select amino acid and vitamin content of the original Eagle’s medium by up to fourfold.

  • Line 274 - MTS kit, what is it, and why is used for? The same for Leave/Dead, this reagent shoud be introduces, whay was used for, how it works.

➢ The MTS (3-(4,5-dimethylthiazol-2-yl)-5-(3-carboxymethoxyphenyl)-2-(4-sulfophenyl)-2H-tetrazolium) assay is used to assess cell proliferation, cell viability and cytotoxicity. The MTS assay protocol is based on the reduction of the MTS tetrazolium compound by viable mammalian cells (and cells from other species) to generate a colored formazan dye that is soluble in cell culture media. This conversion is thought to be carried out by NAD(P)H-dependent dehydrogenase enzymes in metabolically active cells. The formazan dye is quantified by measuring the absorbance at 490-500 nm.

➢ The LIVE/DEAD® Cell Imaging Kit, live cells are distinguished by the presence of ubiquitous intracellular esterase activity as determined by the enzymatic conversion of the virtually non-fluorescent cell-permeant calcein AM to the intensely fluorescent calcein, which is well-retained within live cells. The red component of the LIVE/DEAD® Cell Imaging Kit is cell-impermeant and therefore only enters cells with damaged membranes. In dying and dead cells a bright red fluorescence is generated upon binding to DNA. Background fluorescence levels are inherently low with this assay technique because the dyes are virtually non-fluorescent before interacting with cells. The fluorophores in the LIVE/DEAD® Cell Imaging Kit were selected for their brightness, spectral properties (FITC and Texas Red filters), and ease of use.

  • All these kits in the first paragraph of Methods should be characterized why were used for.

➢ We used 3 kits for determining cell proliferation, cell death, and apoptosis. We explained the purpose of these kits as below.

➢ The MRC5 cells were treated with 10% MTS (3-(4,5-dimethylthiazol-2-yl)-5-(3-carboxymethoxyphenyl)-2-(4-sulfophenyl)-2H-tetrazolium) for determining cell viability and incubated for 1 h at 37°C.

➢ For live/dead cell assay which is distinguished between live and lead cells, MRC5 cells were double-stained with 3 μM Ethidium Homodimer-1 (EthD-1) and 0.3 μM calce-in-AM mixture (Life Technologies, CA, USA) for 30 min in the dark condition.

➢ A FITC Annexin-V Apoptosis Detection Kit I (BD pharmingenTM, La Jolla, CA) was used for determining cellular apoptotic cell death.

  • Authors worked on human cell lines - I do not see any agreements for such research.

➢ MRC-5 (Medical Research Council cell strain 5) is a diploid cell culture line composed of fibroblasts, originally developed from the lung tissue of a 14-week-old aborted Caucasian male fetus. It has been previously reported that heavy metals exposure induces the lung dysfunctions by increasing oxidative stress. There is little evidence regarding the adverse effects of heavy metals in lung fibroblast or lung cancer cells, compared to the numerous studies linking heavy metals to various other cancer types. Therefore, our study on the effects of heavy metals on lung fibroblasts MRC5 cells may help to understand the mechanism of heavy metals in the lung fibroblast. If the reviewer means IRB issue, MRC5 cell lines does not require IRB agreement because it is commercially available thorugh the cell line bank.

  • In my opinion cited literature is not actual, please cite works from last ten years.

➢ In our thought, the original references are important to prove our discovery, so we keep them in the manuscript. However, we also added more references which work with in ten years, as reviewer’s suggestion. The changes are marked as red in the manuscript.

  • How used metals ranges correspond to the environmentsl exposure. Why Authors used such metal concenrations? Is the aim was to detect anormalities, if yes the critic dose was estimated?

➢ The Joint FAO/WHO Expert Committee on Food Additives (JECFA) determined that the tolerable absorption amount of mercury and lead was 3.5 µg/kg and 0.2 µg/kg body weight per day, respectively. To determine the cytotoxicity of HgCl2 and PbAc on human lung fibroblast MRC5 cells, we treated various concentration of HgCl2 (6.25 – 50 µM) and PbAc (20 – 100 µM) for 24 and 48 h in MRC5 cells. The exposure to HgCl2 induced the cell death with 50 µM treatment for 24 and 48 h, whereas the treatment with PbAc has no significant difference on cell viability. In addition, some studies showed that HgCl2 exhibited a significant decrease in cellular viability at 21.3 +/- 6.4 μM (33%; IC50: 65 μM) in Jurkat T cells and 6.44±0.36 to 50.97±19.63 μmol/L in neural cells (Candura et al. 1997; Tong et al. 2016; Karri et al. 2019). Also, PbAc significantly decreased in cellular viability at 30-180 μM  in PC12 cells (Sharifi, Mousavi 2008). Based on these researches and our proliferation assay, we selected the concentration ranges of HgCl2 (12.5 and 50 µM) and PbAc (40 and 100 µM), respectively for the subsequent experiments.

  • Why mercury and lead were used in these particular forms?

➢ Mercury chloride (HgCl2) is a white crystalline substance that is currently used as a catalyst or reagent in various chemical reactions, and to a lesser extent as a disinfectant or pesticide. Mercury exists in various valence states and forms (e.g., He, Hgo, Hg22+, and organic mercury), and upon entering an environmental or biological system in one form (e.g., HgCl2+ in HgCl2) may be changed into a different form with different environmental transport properties, pharmacokinetics and toxicity. The estimated intake of inorganic mercury (including HgCl2) for an adult from air, food, water, and dental amalgam restorations is 3.9 to 24.6 mg Hg2+ per day. In addition, the most mercury has widely spread in the environment as inorganic mercury form (including HgCl2).

➢ Lead acetate is water-soluble and one of the most bioavailable forms of lead. Similar to other lead compounds, it is very poisonous and soluble in water. The commercial form of lead acetate, lead acetate trihydrate, is used as a mordant in textile printing and dyeing, as a lead coating for metals, as a drier in paints, varnishes, and pigment inks, and as a colorant in hair dyes. In this regard, we demonstrated the cytotoxicity of human lung fibroblast cell line, MRC5 by using HgCl2 and PbAc.

  • Line 183 - Authors wrote "recent studies" and gave only 1 reference, please add more ones.

➢ We added the reference in this sentence (Kim et al. 2020).

  • "Even if heavy metals are well studied in various cancers, it is unclear 194 whether heavy metals have carcinogenetic or cytotoxic effects in lung cancer." - very not informative. Authors study 2 metals not all of them.

➢ We replaced the sentence to “Even if heavy metals are well studied in various cancers, it is unclear whether mercury and lead have carcinogenetic or cytotoxic effects in lung cancer.”

  • Figure 6 and 7 must be in results, and when we will do this, we will see that a duscussion part is almost absent. Results that authors received needs to be earnestly discussed with the most actual results involved in most recent literature. Lots od sentences in discussion chapter overlap with very general and not precise sentences from Introduction chapter.

➢ As reviewer’s suggestion, we replaced figure 6 and 7 in result section, and added these results in the discussion section as below. That added paragraph was marked as red text.

➢ “Although the induction of ROS generation by heavy metals was not elucidated in this study, our results showed that mercury increased apoptotic cell death in human lung fibroblast MRC5 cells (Figure 6 and 7). One potential mechanism underlying of cytotoxicity in human fibroblast cell growth is alterations in apoptosis. The activity of caspase-3 was selected as the apoptotic marker because it is a common feature for both caspase-8 (extrinsic) and caspase-9 (intrinsic) activation. Since there are various potential mode of action for HgCl2, it might be one of the best experiments to explain and the change of caspase-8 and caspase-9 activity would cause to alteration in caspase-3 activity. Previous studies have shown that mercury has damaging effects on mitochondria in cerebellar granule cells. In addition, mercury activated caspase-3 pathway which is final apoptotic signaling in differentiated human neurons, astrocyte, and cortex of adult rat. Consistent with the above evidence, our study also proposed that mercury induced apoptosis via increase of caspase-3 activity.”

  • References needs to be enriched with the significant papers to this study.

➢ As reviewer’s suggestion, we added more references to support this report. The changes are marked as red in the manuscript.

Reviewer 2 Report

this work brings rather limited novelty. Eg in Discusion authors are based mostly on the papers already published. Authors present some their own data but lead is not toxic?? in their experimenete or rather slightly toxic. so this part is not excititng. Mercury have some impact on the toxicity of the used cells. English should be checked. for me it is the average level not very usefulk in the field.

Author Response

[Comments and Suggestions for Authors]

This work brings rather limited novelty. Eg in Discusion authors are based mostly on the papers already published. Authors present some their own data but lead is not toxic?? in their experimenete or rather slightly toxic. so this part is not excititng. Mercury have some impact on the toxicity of the used cells. English should be checked. for me it is the average level not very usefulk in the field.

  • We appreciate for reviewer’s comments and suggestions. Because there are various methods to analyze the cytotoxicity of chemical, it is difficult to discover its own function (the mode of action) in various cells. Accordingly, we established the “standard operating procedure (SOP)” for the measuring of cellular physiology in terms of cell viability, proliferation and apoptosis. The results of the present experiments show for the first time that exposure of mercury and lead has an adverse effect on lung fibroblast using our SOP. As reviewer mentioned, this is why previous papers look similar to the present study. However, this study is meaningful to understand the molecular mechanism of each chemical in different cell lines, because each chemical has different toxicological properties in the different cell types. To further investigate the comprehensive effects of mercury and lead, we are now performing genome-wide analysis including RNA-seq, DNA methylation sequencing. It would be helpful to perceive the adverse effects of chemical derivatives and/or analogues in various cell lines in the future studies. We have rephrased the manuscript and gotten second language corrections from professional editors. We also included the certificate of language editing in the cover letter.

Round 2

Reviewer 1 Report

  1. "In the present study, we demonstrated the cytotoxicity of heavy metals by using a human lung fibroblast cell line, MRC5. We investigated the molecular mechanisms by which heavy metals affect the cellular progress of the human lung fibroblast cells. We found that mercury chloride (HgCl2) significantly decrease cell viability of human lung fibroblast cells. In addition, we examined that HgCl2 increased cell cycle arrest at sub-G1 and G2/M phase. Furthermore, the apoptotic cell death was increased via caspase-3 activation in HgCl2 treated MRC5 cells. However, lead acetate (PbAc) has no effect on the development of human lung fibroblast cells. These results indicate that mercury chloride leads to the cell cycle arrest and apoptotic cell death through caspase-3 pathway in human lung fibroblast MRC5 cells." - Dear Authors, in this part of Introduction please explain known/identified/ or possible molecular pathways but please do not include your conclusions, which possess its place in discussion and conclusion at the end of this study. Your paper is focused on molecular pathways, which are not specifically introduced and explained so far in your paper. Again, please write about molecular ways, and clarify the aim of your work, not results!
  2. Another thing related to introduction is that your research is focused on lead acetate. Of course, you are explaining the role of lead. After your last changes lead acetate was deleted from the title, and now the title does not reflect all your work. Maybe you can consider the following title:  Mercury Chloride but not lead acetate causes apoptotic cell death in human lung fibroblast MRC5 cells via regulation of cell cycle progression. Maybe, you have a different idea, but in my opinion without lead acetate the title does not shows all your work.
  3. MRC5 cells were seeded - please use different words.
  4. This should be involved in paragraph 4.3. MTS assay: The MTS assay protocol is based on the reduction of the MTS tetrazolium compound by viable mammalian cells (and cells from other species) to generate a colored formazan dye that is soluble in cell culture media. This conversion is thought to be carried out by NAD(P)H-dependent dehydrogenase enzymes in metabolically active cells. The formazan dye is quantified by measuring the absorbance at 490-500 nm using a Multiskan GO microplate reader 323 (Waltham, MA).
  5. "For live/dead cell assay which is distingushed between live and dead cells ...". After the paragraph involving this sentence the explanation from the part for Reviewers should be added: The LIVE/DEAD® Cell Imaging Kit, live cells are distinguished by the presence of ubiquitous intracellular esterase activity as determined by the enzymatic conversion of the virtually non-fluorescent cell-permeant calcein AM to the intensely fluorescent calcein, which is well-retained within live cells. The red component of the LIVE/DEAD® Cell Imaging Kit is cell-impermeant and therefore only enters cells with damaged membranes. In dying and dead cells a bright red fluorescence is generated upon binding to DNA. Background fluorescence levels are inherently low with this assay technique because the dyes are virtually non-fluorescent before interacting with cells. The fluorophores in the LIVE/DEAD® Cell Imaging Kit were selected for their brightness, spectral properties (FITC and Texas Red filters), and ease of use.
  6. In the whole text a letter X, for example in 1X PBS, should be replaced by a symbol.
  7. "an MTS assay" - instead an should be a
  8. In line 82 please explain why such doses were chosen (material for the answer to reviewer). What the literature says?
  9. "Proliferating cells express high amounts of Ki-67" - please explain the Ki-67 in Introduction
  10. Lines 230-251 (Discussion part) includes basic informations, which are mentioned in Introduction. These informations do not bring new data to discuss the received results.
  11. "Since there are various potential mode of action for HgCl2, it might be one of the best experiments to explain and the change of caspase-8 and caspase-9 activity would cause to alteration in caspase-3 activity." Rewrite this sentence, please.
  12. "Therefore, future studies will determine whether ... " in Conclusion - will help to determine. It is difficult to predict that a future experiment will for sure bring expected result, in my opinion.
  13. If Authors will provide more informative molecular introduction (cell cycle, apoptosis, and the role of enzymes), I am sure they will enlarge their reference list.

Reviewer 2 Report

revision make thispaper more clear so it could be considered for publication.
